# Tensor Decomposition Based Memory-Efficient Incremental Learning

Yuhang Li [1]  Guoxu Zhou [1 2]  Zhenhao Huang [1]  Xinqi Chen [1]  Yuning Qiu [3]  Qibin Zhao [3 1]

## Abstract

Class-Incremental Learning (CIL) has gained considerable attention due to its capacity to accommodate new classes during learning. Replay-based methods demonstrate state-of-the-art performance in CIL but suffer from high memory consumption to save a set of old exemplars for revisiting. To address this challenge, many memory-efficient replay methods have been developed by exploiting image compression techniques. However, the gains are often bittersweet when pixel-level compression methods are used. Here, we present a simple yet efficient approach that employs tensor decomposition to address these limitations. This method fully exploits the low intrinsic dimensionality and pixel correlation of images to achieve high compression efficiency while preserving sufficient discriminative information, significantly enhancing performance. We also introduce a hybrid exemplar selection strategy to improve the representativeness and diversity of stored exemplars. Extensive experiments across datasets with varying resolutions consistently demonstrate that our approach substantially boosts the performance of baseline methods, showcasing strong generalization and robustness.

## 1. Introduction

In recent years, deep learning has achieved remarkable advancements, with neural networks reaching or even surpassing human-level performance in many domains (Silver et al., 2016; Jumper et al., 2021). Traditionally, deep networks rely heavily on offline training, requiring multiple epochs over a static, pre-collected dataset. However, in the open world, data is continuously generated and constantly changing (Gomes et al., 2017), resulting in a consistent delay in incorporating new knowledge by those advanced models (e.g., large language models such as GPT-4 (Achiam et al., 2023)). To address this, researchers have turned to a promising approach known as "Class-Incremental Learning" (CIL) (Masana et al., 2022; Zhou et al., 2023b). CIL aims to learn from streaming data continually without forgetting previously learned. Despite this vision, when the model directly trains on new data, it tends to forget a substantial amount of previously captured knowledge, leading to irreversible performance degradation—a phenomenon known as catastrophic forgetting (McCloskey & Cohen, 1989; Robins, 1993). Therefore, effectively combating catastrophic forgetting is the central challenge in developing CIL methods.

Replay-based methods (Rebuffi et al., 2017; Lopez-Paz & Ranzato, 2017; Isele & Cosgun, 2018; Zhao et al., 2020; Bang et al., 2021; Wang et al., 2022a), inspired by human learning processes (Robins, 1993; Kumaran et al., 2016; Kudithipudi et al., 2022), have demonstrated state-of-the-art performance in various incremental learning scenarios. These methods explicitly store a subset of previously encountered data (exemplars) in memory. When training on new tasks, the model can revisit these exemplars to help mitigate catastrophic forgetting. While intuition suggests that storing more old exemplars could enhance performance, practical constraints, e.g., fixed memory capacity (Krempl et al., 2014; Rebuffi et al., 2017), restrict only a few data can be kept for replay. Consequently, there is a severe imbalance between new and old data, with the training process always dominated by new ones.

To alleviate the above problem, various **Memory-Efficient Replay Methods** (Wang et al., 2024) have been proposed. For instance, MECIL (Zhao et al., 2021) proposes using low-fidelity samples instead of the original ones to transfer old class knowledge. CIM (Luo et al., 2023) suggested downsampling non-discriminative pixels (e.g., background) while retaining discriminative pixels (e.g., foreground) in the original image. MAE-CIL (Zhai et al., 2023) demonstrates that Masked Autoencoders (MAE) (He et al., 2022) are efficient incremental learners, which store random image patches as exemplars and they can reconstruct high-quality images from only partial information for replay. Although

[1]School of Automation, Guangdong University of Technology, Guangzhou, CHINA [2]Key Laboratory of Intelligent Detection and the Internet of Things in Manufacturing, Ministry of Education, Guangdong University of Technology, Guangzhou, China [3]RIKEN AIP, Tokyo, JAPAN. Correspondence to: Guoxu Zhou <gx.zhou@gdut.edu.cn>.

*Proceedings of the 42st International Conference on Machine Learning*, Vancouver, Canada. PMLR 267, 2025. Copyright 2025 by the author(s).

these methods achieve notable performance, they struggle to reconcile data-friendliness and model-friendliness (see Sec. 2 for more details).

Pixel-level compression methods, such as MECIL and CIM, directly compress images in the original high-dimensional pixel space, overlooking the fact that natural images typically have low intrinsic dimensionality (Levina & Bickel, 2004) and exhibit local connectivity. To illustrate, in image reconstruction tasks, Masked Autoencoder (He et al., 2022) divides an image into multiple patches and reconstructs the complete image by randomly masking a certain proportion of these patches. Experimental results show that with 75% of the patches masked, the original image can still be reconstructed. Similarly, low-rank tensor decomposition-based completion methods can recover images with high fidelity, even when over 80% of their data is missing (Yokota et al., 2016; Qiu et al., 2024). Additionally, the Maximum Likelihood Estimation of the intrinsic dimensionality of images on the ImageNet (Deng et al., 2009) indicates that, although each image contains 150,528 pixels, its intrinsic dimension is only between 26 and 43 (Levina & Bickel, 2004). These results suggest that compressing data directly in high-dimensional pixel space is inefficient, and it can be represented or approximated with much less complexity.

Building on these insights, we propose introducing low-rank tensor decomposition (TD) (Kolda & Bader, 2009) for image compression. This method takes full advantage of natural images' local correlation and low intrinsic dimensionality. Not only can it achieve considerable compression efficiency, but it also effectively captures the internal multi-dimensional structure of the image (e.g., the spatial distribution of pixels and the correlation between channels) (Kilmer et al., 2021) so that more discriminative information can be retained after compressed, which is an attribute crucial for memory-efficient replay methods. Furthermore, recognizing the importance of balancing the quality and quantity of exemplars for this kind of method (Zhao et al., 2021; Wang et al., 2022b), we propose a novel exemplar selection strategy to enhance the representativeness and diversity of the exemplars.

To evaluate the proposed method, we integrate it into several advanced CIL methods (i.e., iCaRL (Rebuffi et al., 2017), FOSTER (Wang et al., 2022a), DER (Yan et al., 2021), MOME (Zhou et al., 2022)) and perform extensive experiments on different-scale datasets. The results demonstrate that our proposed method consistently delivers substantial improvements. To summarize, our contributions are as follows: 1) We propose a memory-efficient CIL method based on TD. To the best of our knowledge, tensor methods have not been used in CIL to mitigate catastrophic forgetting. Our work represents a new attempt, and we anticipate that introducing the tensor approach will yield valuable insights and

solutions to the challenges posed by CIL. 2) Distinguished from most existing memory-efficient methods, our approach can be effectively applied to data of various resolutions and easily integrated with other rehearsal methods as a plug-and-play solution, serving as a model-friendly and data-friendly method. 3) In the context of memory-efficient CIL methods, we introduce an innovative exemplar selection strategy that effectively balances the quantity and quality of samples and improves performance to some extent. This strategy is adaptable and can be generalized to other similar methods. 4) Extensive experiments consistently indicate that the proposed method achieves excellent performance across diverse experimental settings while exhibiting low sensitivity to parameter variations, underscoring the method's strong generalization and robustness.

## 2. Related Works

There are three principal strategies to mitigate catastrophic forgetting in incremental learning (Van de Ven et al., 2022; Zhou et al., 2023b). **Regularization-based Methods** (Kirkpatrick et al., 2017; Wang et al., 2022c) introduce additional regularization terms into the loss function to penalize significant changes in the weights crucial for previously learned tasks. This ensures that the model retains its performance on old tasks while learning new ones. **Architecture-based Methods** (Yan et al., 2021; Zhou et al., 2022) aims to mitigate catastrophic forgetting by dynamically modifying the network architecture. A fundamental concept here is to preserve the knowledge of past tasks by freezing the parameters of previously trained sub-networks (called "old model"). New network structures are then introduced to accommodate new knowledge from incoming tasks. **Replay-based Methods** (Rebuffi et al., 2017; Lopez-Paz & Ranzato, 2017; Wang et al., 2022a) maintain a memory buffer with a fixed budget to store a few examples from previous tasks. When training on new tasks, the model can revisit these examples to reinforce its memory of old knowledge, thereby preventing forgetting.

Beyond this, some **Memory-Efficient Replay Methods** seek to improve storage efficiency by compressing raw images or storing feature-level data, achieving impressive performance. Works centred on feature storage (Iscen et al., 2020; Hayes et al., 2020; Toldo & Ozay, 2022), referred to as feature replay (Wang et al., 2024), offer considerable efficiency and privacy benefits by preserving feature-level distributions rather than raw data. However, a major hurdle is representation drift resulting from the feature extractor's sequential updating, a phenomenon indicative of feature-level catastrophic forgetting (Wang et al., 2024). Knowledge Distillation (KD) (Hinton et al., 2015) and the partially fixed feature extractor (Hayes et al., 2020) are compromised solutions for this problem.

Another work line improves storage efficiency by compressing images (Caccia et al., 2020; Wang et al., 2022b; Luo et al., 2023; Zhai et al., 2023). For example, MECIL (Zhao et al., 2021) employs an additional auto-encoder to transform high-fidelity images into lower-fidelity ones and introduces an adaptive training scheme to mitigate performance degradation caused by domain shifts. CIM (Luo et al., 2023) suggests downsampling non-discriminative pixels (e.g., background) while retaining discriminative pixels (e.g., foreground) in the original image and introduces additional Paué Activation Units (PAU) (Molina et al., 2019) to determine the activation areas at each incremental stage dynamically, but this method is not efficient for low-resolution datasets, such as CIFAR (Krizhevsky, 2009). MAE-CIL (Zhai et al., 2023) stores random image patches as exemplars, and they can reconstruct high-quality images from only partial information for replay. This method achieved state-of-the-art performance, but the reconstruction heavily depends on MAE (He et al., 2022).

Although these methods achieve notable performance, they necessitate the introduction of well-crafted additional structures or strategies to compensate for the negative impacts of compression. In contrast, tensor decomposition-based compression methods characterize the image using a set of factors with fewer parameters. These factors are stored and can be reconstructed during subsequent task training. Thanks to the images' low-rank property, the reconstruction error remains minimal, ensuring that the reconstructed images retain the same quality as the originals. This method serves as a training plugin, significantly boosting the performance of existing replay-based CIL techniques.

## 3. Methodology

This section introduces the basic concepts and notations, presents the classical incremental learning setup, and then describes how our proposed **T**ensor **D**ecomposition-based **M**emory-**E**fficient **R**eplay (**TDMER**) works. Fig. 2 illustrates the framework of our methods.

### 3.1. Preliminary

**Incremental Learning** aims to train a model on a sequence of tasks, represented by their datasets $\{\mathcal{D}^1, \mathcal{D}^2, \cdots, \mathcal{D}^T\}$, and achieve good performance on unseen test data from those tasks. The training set of task $t$ contains $N_t$ distinct data-label pairs, $\mathcal{D}^t = \{(x_n^t, y_n^t)\}_{n=1}^{N_t}$, where $x_n^t \in X_t$ and $y_n^t \in Y_t$ represent the input instance and their corresponding label, respectively. The goal is to let the model, parameterized by $\theta$, learn a mapping from the whole input space $\mathcal{X}_T = \bigcup_{t=1}^T X_t$ to the label space $\mathcal{Y} = \bigcup_{t=1}^T Y_t$, i.e. $f_\theta : \mathcal{X} \to \mathcal{Y}$. At the test stage, the model should be able to predict the correct label $y$ for a new unseen sample $x$ drawn from any of the previous tasks.

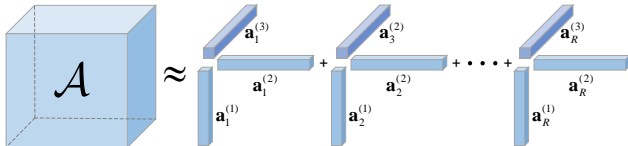

*Figure 1.* Illustration of a CP decomposition of $third$-order tensor $\mathcal{A}$ into a sum of rank-1 tensors

There are three common kinds of incremental learning scenarios: Domain-incremental learning (DIL), Task-Incremental Learning (TIL), and Class-Incremental Learning (CIL) (Van de Ven et al., 2022; Zhou et al., 2023b). In the DIL setting, all tasks share the same label space, the training data distributions for different tasks may differ (e.g., various image styles), and the model does not receive information about which task the test sample belongs to during inference. For TIL and CIL settings, the label spaces for different tasks are disjoint; the model receives information about task identity during testing for TIL but not for CIL. In this paper, we focus on the more representative and challenging CIL.

**Tensor Decomposition** is a mathematical technique for analyzing multidimensional data represented by tensors (Kolda & Bader, 2009). It breaks down a complex, higher-order tensor into a combination of simpler, lower-order tensors. Here, we briefly introduce the CANDECOMP/PARAFAC (CP) (Carroll & Chang, 1970) decomposition for an $N^{th}$-order tensor $\mathcal{A} \in \mathbb{R}^{I_1 \times I_2 \times \cdots \times I_N}$. Formally, the CP decomposition decomposes $\mathcal{A}$ into a sum of $R$ rank-one factor tensors:

$$\mathcal{A} = \sum_{r=1}^R \mathbf{a}_r^{(1)} \circ \mathbf{a}_r^{(2)} \circ \cdots \circ \mathbf{a}_r^{(N)} \tag{1}$$
$$= [\![\mathbf{A}^{(1)}, \ldots, \mathbf{A}^{(N)}]\!],$$

where rank $R$ is a predefined positive integer, the symbol "$\circ$" represents the vector outer product, and $\mathbf{a}_r^{(1)} \in \mathbb{R}^{I_1}, \mathbf{a}_r^{(2)} \in \mathbb{R}^{I_2}, \ldots, \mathbf{a}_r^{(N)} \in \mathbb{R}^{I_N}$ for $r = 1, \ldots, R$. These vectors can be grouped into $N$ matrices, i.e. $\{\mathbf{A}^{(n)} \in \mathbb{R}^{I_n \times R}\}_{n=1}^N$, where $\mathbf{A}^{(n)} = [\mathbf{a}_1^{(n)}, \mathbf{a}_2^{(n)}, \cdots, \mathbf{a}_R^{(n)}]$ for $n = 1, \ldots, N$, and $[\![\cdot]\!]$ is defined as the CP decomposition operator (Kolda & Bader, 2009). Elementwise, the CP decomposition is written as:

$$a_{i_1, i_2, \ldots, i_N} \approx \sum_{r=1}^R \mathbf{a}_r^{(i_1)} \mathbf{a}_r^{(i_2)} \cdots \mathbf{a}_r^{(i_N)}. \tag{2}$$

For better clarity, a schematic representation of the CP decomposition of a $third$-order tensor is shown in Fig. 1

### 3.2. TD-Based Memory-Efficient Replay Method.

We introduce TD methods to factorize the images into a set of factor matrices or tensors. By selecting an appro-

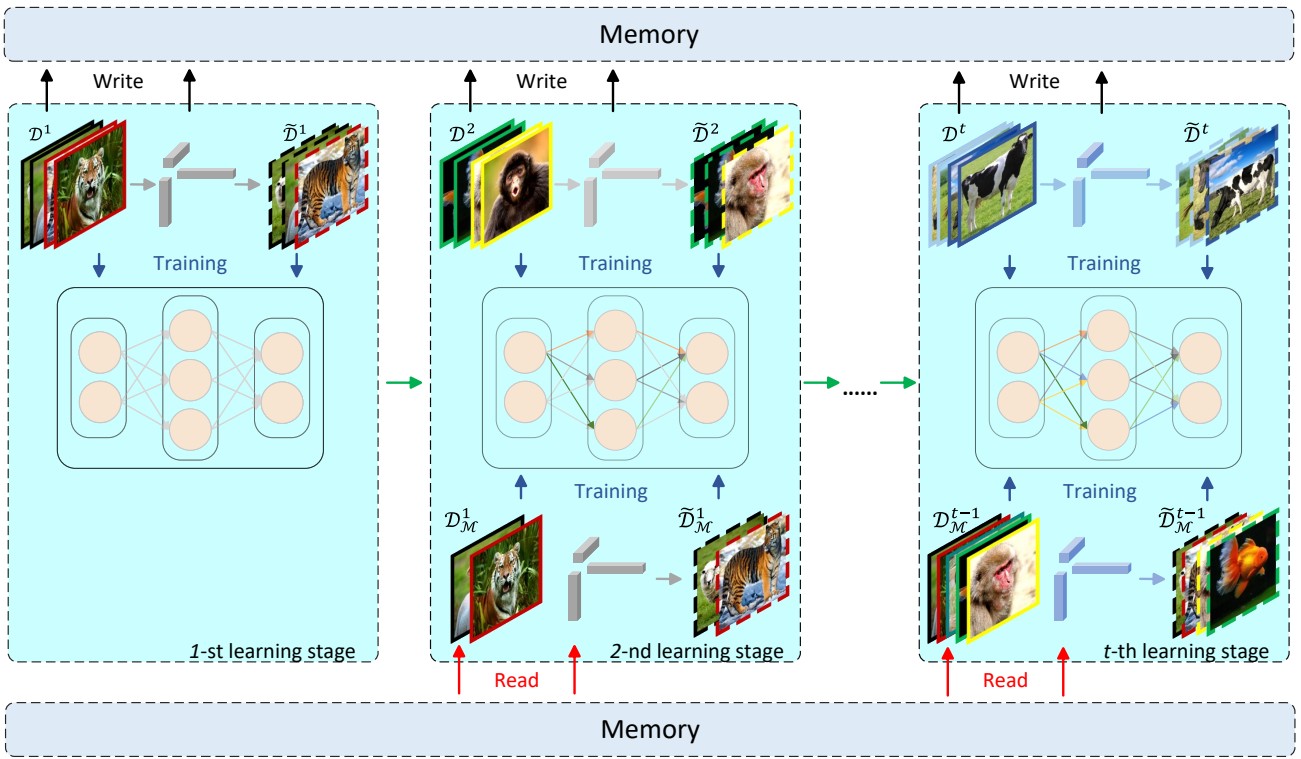

*Figure 2.* Illustration of the proposed tensor decomposition-based memory-efficient CIL method with two-stage exemplar selected strategy. We use $\mathcal{D}^t$ to represent the training data of $t$-th task, $\tilde{\mathcal{D}}^t$ represent the partial reconstructed data of $\mathcal{D}^t$, $\mathcal{D}^t_M$ represent the exemplar stored in memory, and $\tilde{\mathcal{D}}^t_M$ represent the exemplar reconstructed by the stored factor. The notion of "Write", "Read," and "Training" refers to writing raw samples and factors obtained from other sample decomposition into memory, reading exemplars and factors set from memory, and training the model using new task data, stored exemplars, and reconstructed samples from the factors, respectively. All these exemplars are chosen by our exemplars selection strategy.

priate rank (crucial for balancing the fidelity and compactness of the factors), we can significantly reduce the size of these components while preserving crucial information. We first investigate two widely used TD methods: CANDECOMP/PARAFAC (Carroll & Chang, 1970) and Tucker (Tucker, 1966). However, maintaining decomposition accuracy with Tucker decomposition results in component sizes close to the original data for low-resolution datasets such as CIFAR-10/100 (Krizhevsky, 2009), which are not effectively handled by many memory-efficient replay methods (Wang et al., 2022c; Luo et al., 2023). This is not desirable for our purposes. To ensure generalizability, we chose to use CP decomposition exclusively.

**From CP Decomposition to Image Compression.** For the data $x^t_n \in \mathbb{R}^{C \times H \times W}$ in $\mathcal{D}^t$ , given a rank $R$, the CP decomposition factorizes it into a sum of $R$ rank-1 tensor factors. According to Eq. (1), we can infer the size of these components (be grouped into matrices) is $\mathbf{A}^{(1)} \in \mathbb{R}^{C \times R}$, $\mathbf{A}^{(2)} \in \mathbb{R}^{H \times R}$, $\mathbf{A}^{(3)} \in \mathbb{R}^{W \times R}$, respectively. Therefore, the compression ratio $\eta$ of this process can be defined as

follows:

$$\eta = \frac{(C + H + W) \times R}{C \times H \times W}. \quad (3)$$

Note that with Tensorly (Kossaifi et al., 2019) for tensor decomposition, we first transform the original image (in integer format) to float-point format. As a result, the factors from the decomposition are in floating-point format. To ensure the Eq. (3) holds, we need to convert these float-point factors to 8-bit. By doing this, we can consistently achieve a compression ratio of less than 1, where 1 represents the memory required to store the raw image. Therefore, within the same memory budget, we can store more decomposed factors.

**Adaptive Training.** Following the above concepts outlined, we focus on the CIL setting, where for all $t \neq t'$, $\mathcal{X}_t \cap \mathcal{X}_{t'} = \varnothing$ and $\mathcal{Y}_t \cap \mathcal{Y}_{t'} = \varnothing$. The goal is to conduct continuous training without forgetting previous knowledge. Specifically, for conditioning the current task $t$, the setting with CIL remains rather tricky. The model must maintain stability for retaining the knowledge from all previous tasks

$(1, 2, \ldots, t-1)$, also requiring continuous learning to adapt to the new task $t$. This results in the stability-plasticity dilemma. To overcome this bottleneck, replay-based methods employ a small-sized memory buffer, denoted as $\mathcal{D}_M$. Old tasks' exemplars are sampled, stored, and updated after completing training on each task. Data from the current task $t$ is presented along with the exemplar set to the model. Thus, the objective function for replay-based CIL methods can be written as:

$$
\begin{aligned}
&\arg \min_\theta \sum_{(x,y) \in \mathcal{D}^t} l\left(f_\theta(x), y\right) + \lambda \mathcal{L}_{reg}, \\
&\mathcal{L}_{reg} := \sum_{(x,y) \in D_M^{t-1}} l_{reg}\left(f_\theta(x), y\right),
\end{aligned}
\tag{4}
$$

where $l_{reg}$ denotes the regularization terms based on the old exemplars, and $\lambda$ is a hyperparameter controlling the strength of the regularization.

Note that both the tensor decomposition and the data type conversion introduce some minor errors. To mitigate these effects, we adopt an adaptive training strategy. The objective function is as follows:

$$
\arg \min_\theta \sum_{(x,y) \in \mathcal{D}^t \bigcup \tilde{\mathcal{D}}^t} l\left(f_\theta(x), y\right) + \lambda \mathcal{L}_{reg}.
\tag{5}
$$

Namely, in each training epoch, we select a subset of $D^t$, denoted as $\{x_1^t, x_2^t, \cdots, x_N^t\}$, and decompose each data to low-rank factors, which can be reconstructed as $\{\tilde{x}_1^t, \tilde{x}_2^t, \cdots, \tilde{x}_N^t\} = \tilde{\mathcal{D}}^t$. This allows the model to emulate training with processed images and learn to overcome the effects of error.

**Exemplar Selection Strategy.** We notice that existing methods almost always keep only compressed samples, but not all compressed samples have a positive effect; on the contrary, some have a negative impact (as shown in Fig. 3). It means that emphasizing quantity at the expense of quality is counterproductive in memory-efficient CIL. To achieve a better trade-off, we propose a two-step hybrid sample selection strategy.

Specifically, in the first step, we use a common strategy called herding (Rebuffi et al., 2017) to select a part of raw data $\{x_1^t, x_2^t, \cdots, x_i^t\}$, denoted as $\mathcal{D}_M^{t-1}$. Herding helps identify and choose the samples that best represent the characteristics of their respective classes, meaning focus on quality.

In the second step, we randomly select a portion of the samples that were not chosen in the first stage, e.g. $\hat{D}_M^{t-1} = \{\hat{x}_1^t, \hat{x}_2^t, \cdots, \hat{x}_j^t\}$, and $\mathcal{D}_M^{t-1} \cap \hat{D}_M^{t-1} = \varnothing$, get their decomposition factors and store them, denoted as $\tilde{\mathcal{D}}_M^{t-1}$. In the training, factors in $\tilde{\mathcal{D}}_M^{t-1}$ can be reconstructed flexibly, which denoted as $\{\tilde{x}_1^t, \tilde{x}_2^t, \cdots, \tilde{x}_j^t\}$. We use relative error to evaluate the quality of reconstruction, which is denoted as:

$$
RE_j = \frac{||\hat{x}_j^t - \tilde{x}_j^t||_F^2}{||\hat{x}_j^t||_F^2}.
\tag{6}
$$

To ensure the quality of the reconstructed data, we introduce a hard threshold $\tau$; if the reconstruction error is larger than this threshold, which means that most of the critical information may have been lost, this data is discarded, and another one is re-selected. Thus, the final objective function is formulated as:

$$
\begin{aligned}
&\arg \min_\theta \sum_{(x,y) \in \mathcal{D}^t \bigcup \tilde{\mathcal{D}}^t} [l\left(f_\theta(x), y\right)] + \lambda_1 \mathcal{L}_{reg} + \lambda_2 \tilde{\mathcal{L}}_{reg}, \\
&\tilde{\mathcal{L}}_{reg} := \sum_{(x,y) \in \tilde{D}_M^{t-1}} l_{reg}\left(f_\theta(x), y\right),
\end{aligned}
\tag{7}
$$

where $\lambda_1$ and $\lambda_2$ is the hyperparameter making a trade-off between $\mathcal{D}_M^{t-1}$ and $\tilde{\mathcal{D}}_M^{t-1}$, $l_{reg}$ means the same regularization function.

We introduce a parameter $\epsilon$ to control the proportion of samples between these two stages. When $\epsilon$ is equal to 0, which means that we only use the exemplars in $\tilde{\mathcal{D}}_M^{t-1}$, and $\lambda_1$ is set to 0; when $\epsilon$ is equal to 1, it means that we only use the exemplars in $\mathcal{D}_M^{t-1}$, and $\lambda_2$ is equal to 0. Notably, the number of samples selected in the first stage typically constitutes only a modest part, meaning a focus on quantity.

## 4. Experiments

### 4.1. Experimental Settings

**Datasets.** We conducted experiments on two widely used datasets. CIFAR-100 (Krizhevsky, 2009) is a low-resolution and widely used dataset for image classification tasks. It comprises 60,000 32×32 RGB images categorized into 100 classes, each containing 500 training and 100 testing images. ImageNet-100 (Deng et al., 2009) is a subset of the larger ImageNet-1000 containing 100 randomly selected categories. Each category has approximately 1,300 training samples and 50 test samples, all high-resolution but varying in spatial size.

**Protocols.** We conduct our experiments following two protocols: learning from scratch (LFS) and learning from half (LFH), which are two different ways to split the classes into incremental phases. **LFS**: divides all the classes equally in each incremental stage. For example, if there are $M$ stages and $N$ classes, each task has $N/M$ classes for training. **LFH**: the first increment task divides half of the total classes, and the rest are equally divided among the subsequent stages. Namely, the $N/2$ classes are distributed as the first task, and the $N/2(M-1)$ classes are distributed as the remaining tasks.

*Table 1.* average accuracy (%) and last accuracy (%) of four baseline methods on CIFAR-100, with or without our TDMER plugged-in, the memory budget $M = 2k$.

| Method | Base0 Inc10 | | Base0 Inc20 | | Base50 Inc10 | |
| --- | --- | --- | --- | --- | --- | --- |
| | Avg | Last | Avg | Last | Avg | Last |
| iCaRL | 60.83 | 42.99 | 62.85 | 46.67 | 55.99 | 46.67 |
| iCaRL w/ ours | 67.22 (+6.39) | 51.98 (+8.99) | 68.05 (+5.20) | 56.07 (+9.40) | 61.59 (+5.60) | 53.37 (+6.70) |
| FOSTER | 66.67 | 54.24 | 69.50 | 60.11 | 66.75 | 58.76 |
| FOSTER w/ ours | 70.61 (+3.94) | 58.25 (+4.01) | 72.16 (+2.66) | 62.41 (+2.30) | 69.17 (+2.42) | 61.79 (+3.03) |
| MEMO | 69.57 | 58.13 | 69.61 | 60.60 | 65.95 | 58.45 |
| MEMO w/ ours | 72.67 (+3.10) | 62.44 (+4.31) | 72.47 (+2.86) | 64.40 (+3.80) | 68.61 (+2.66) | 63.28 (+4.83) |
| DER | 71.05 | 59.64 | 71.14 | 63.17 | 69.37 | 62.71 |
| DER w/ ours | **73.69** (+2.64) | **63.21** (+3.57) | **73.57** (+2.43) | **66.14** (+2.97) | **70.79** (+1.42) | **65.16** (+2.45) |

Both protocols are widely adopted in the current CIL community (Yan et al., 2021; Wang et al., 2022a; Zhou et al., 2022; Luo et al., 2023). To better distinguish, we use '**Base-**$b$, **Inc-**$c$' to denote different data splits, where $b$ represents the number of classes in the first stage and $c$ represents the number of courses in each incremental task. For example, "Base 0, Inc 10" indicates that the LFS protocol is followed, and ten classes are assigned to each incremental task.

**Baseline Methods.** We incorporate our method into four baseline CIL methods, and they are listed as: **iCaRL** (Rebuffi et al., 2017) combines knowledge distillation and prototype rehearsal with several innovative components, i.e., nearest-mean-of-exemplars classifier and prioritized exemplar selection; **DER** (Yan et al., 2021) first freeze previously learned representation and expand a new feature extractor when new tasks come, then use a linear layer to aggregate the feature; **MEMO** (Zhou et al., 2022) proposes extending only the deeper layers specific to the task, sharing a more generalized shallow network for enabling model expansion with the least budget cost; **FOSTER** (Wang et al., 2022a) utilize a new module to fit the residuals between the target and the output of the original model, after which it removes redundant parameters and feature dimensions via a performant distillation strategy to maintain the single backbone model.

**Evaluation Metrics.** We use two metrics to evaluate the performance. First, *Last Accuracy* (Last) is used to estimate the final performance of CIL models, which calculates the classification accuracy for all seen classes up to the current task. However, reporting only the final accuracy ignores performance varying over the entire learning process. Hence, we employ *Average Accuracy* (Avg) to reflect the historical performance, which is defined as $AA_t = \frac{1}{t}\sum_{i=1}^{t} a_{t,i}$, where $a_{t,i} \in [0, 1]$ denotes the accuracy of task $i$ after learning task $t$ $(i \leq t)$. We shall report the final average accuracy, i.e., $AA_T$. This metric considers the performance over the entire learning trajectory. Note that we only report the top-1 accuracy in all experimental results.

*Table 2.* average accuracy (%) and last accuracy (%) on CIFAR-100 of MEMO and DER with or without our TDMER integrated, under two memory budgets, $M = 1k$, $M = 500$.

| Budget | Method | Base0 Inc10 | |
| --- | --- | --- | --- |
| | | Avg | Last |
| $M = 500$ | MEMO | 60.53 | 45.67 |
| | MEMO w/ ours | **67.61** | **55.76** |
| | DER | 64.74 | 50.95 |
| | DER w/ ours | **69.47** | **56.89** |
| $M = 1k$ | MEMO | 65.15 | 52.08 |
| | MEMO w/ ours | **70.51** | **59.30** |
| | DER | 68.50 | 55.86 |
| | DER w/ ours | **71.57** | **61.02** |

**Implementation Details.** Our implementation is based on the deep learning library PyTorch (Paszke et al., 2019), the CIL toolbox PyCIL (Zhou et al., 2023a), and the tensor development library Tensorly (Kossaifi et al., 2019). Following (Yan et al., 2021; Wang et al., 2022a; Zhou et al., 2022; Luo et al., 2023), we employ the same backbone, i.e., 18-layer ResNet (He et al., 2016) for ImageNet-100 and 32-layer ResNet for CIFAR-100, with a fully-connected layer as the classifier across all our experiments, and all training hyperparameters remain the same with PyCIL. More details will be provided in the appendix. A.

### 4.2. Results and Analyses

We evaluate the performance of our proposed method integrated into four baseline methods on CIFAR-100 and ImageNet-100. More results will be provided in the appendix. B.

**Results on CIFAR-100.** From Tab.1, we have the following observations: (1) Our TDMER, when integrated into the four baseline methods, substantially improves both last accuracy and average accuracy across different data splits.

*Table 3.* average accuracy (%) and last accuracy (%) of four baseline methods on ImageNet-100, with or without our TDMER plugged-in, the memory budget $M = 2k$.

| Method | Base0 Inc10 | | Base0 Inc20 | | Base50 Inc10 | |
|---|---|---|---|---|---|---|
| | Avg | Last | Avg | Last | Avg | Last |
| iCaRL | 60.24 | 41.40 | 66.83 | 50.12 | 55.20 | 43.32 |
| iCaRL w/ ours | 64.38 (+4.14) | 47.34 (+5.94) | 69.38 (+2.55) | 54.40 (+4.28) | 58.80 (+3.60) | 47.12 (+3.80) |
| FOSTER | 69.91 | 60.60 | 76.57 | 69.74 | 70.07 | 63.85 |
| FOSTER w/ ours | 74.51 (+4.60) | 66.40 (+5.80) | 78.59 (+2.02) | 72.30 (+2.56) | 75.05 (+4.98) | 69.58 (+5.73) |
| MEMO | 72.41 | 62.56 | 76.14 | 68.42 | 70.09 | 63.42 |
| MEMO w/ ours | 74.97 (+2.56) | 66.94 (+4.38) | 77.21 (+1.07) | 71.10 (+2.68) | 70.63 (+0.54) | 65.38 (+1.96) |
| DER | 77.08 | 66.84 | 78.56 | 72.10 | 77.57 | 71.10 |
| DER w/ ours | **79.75** (+2.67) | **73.26** (+6.42) | **79.69** (+1.13) | **73.90** (+1.80) | **79.96** (+2.39) | **75.46** (+4.36) |

*Table 4.* average accuracy (%) and last accuracy (%) on ImageNet-100 of MEMO and DER with and without our TDMER integrated, under two memory budgets, $M = 1k$, $M = 500$.

| Budget | Method | Base0 Inc10 | |
|---|---|---|---|
| | | Avg | Last |
| $M = 500$ | MEMO | 63.78 | 50.68 |
| | MEMO w/ ours | **67.64** | **57.66** |
| | DER | 69.86 | 55.86 |
| | DER w/ ours | **72.66** | **62.54** |
| $M = 1k$ | MEMO | 68.99 | 57.92 |
| | MEMO w/ ours | **71.77** | **62.26** |
| | DER | 72.41 | 62.22 |
| | DER w/ ours | **74.95** | **65.36** |

*Table 5.* Comparison of our method with other memory-efficient replay methods on the ImageNet-100 under the 'Base0 Inc10' setting.

| Budget | Method | Base0 Inc10 | |
|---|---|---|---|
| | | Avg | Last |
| $M = 2k$ | MRDC (ICLR'22) | 76.02 | - |
| | CIM (CVPR'23) | 77.94 | - |
| | MAE-CIL (ICCV'23) | 79.54 | 70.29 |
| | Ours | **79.75** | **73.26** |

Specifically, the highest observed increases are 6.39 % for average accuracy and 8.99 % for last accuracy under 10-task setting; (2) Although our improvements are significant across all baseline methods, the extent of enhancement varies, with the largest gains observed for the iCaRL baseline, while the performance boost for the already high-performing DER is comparatively smaller; (3) Furthermore, as the number of tasks increases, the improvements provided by our method also grow, indicating that our approach becomes increasingly beneficial in more complex incremental learning scenarios.

Tab. 2 shows the results of MEMO and DER for two memory budgets on the CIFAR-100. We can see that our method consistently boosts MEMO and DER, achieving higher performance growth with a tighter memory budget. Specifically, it improves the last accuracy of MEMO by 10.09% when $M = 500$, with an increment of 7.28% when $M = 1k$ (the increment is 4.31% when $M = 2k$). DER also shows similar results, but with a slightly smaller increase compared to MEMO, namely, for the three memory budgets $M = 500$, $M = 1k$, and $M = 2k$, the increments are 5.94%, 5.16%, and 3.57%, respectively.

**Results on ImageNet-100.** Keeping the same experimental settings as CIFAR-100, we report the best average accuracy and last accuracy on ImageNet-100 in Tab.3, and Tab. 4 shows the average accuracy and last accuracy of MEMO and DER for two memory budgets on the ImageNet-100. The experimental results on ImageNet-100 corroborate the findings from CIFAR-100. Our proposed method consistently improves the performance of various baseline methods, with the magnitude of improvement varying across methods, increasing with the number of tasks and achieving higher improvement on a stricter memory budget. These consistent trends across different datasets underscore the robustness and effectiveness of our approach.

**Comparing with other efficient replay methods.** We compare our method with a variety of memory-efficient replay methods, such as MRDC (Wang et al., 2022b), CIM (Luo et al., 2023), and MAE-CIL (Zhai et al., 2023). All these methods use the same amount of storage, i.e., 20 original images per class. Tab. 5 shows the results on the ImageNet-100 under the 'Base0 Inc10' setting. Our method achieves consistently higher performance.

**Effectiveness of Compression Methods.** Memory-efficient replay methods improve storage efficiency by retaining more compressed data, and the quality of the compressed samples is critical to whether such methods work. To evaluate the compression techniques, we processed the CIFAR-100 train-

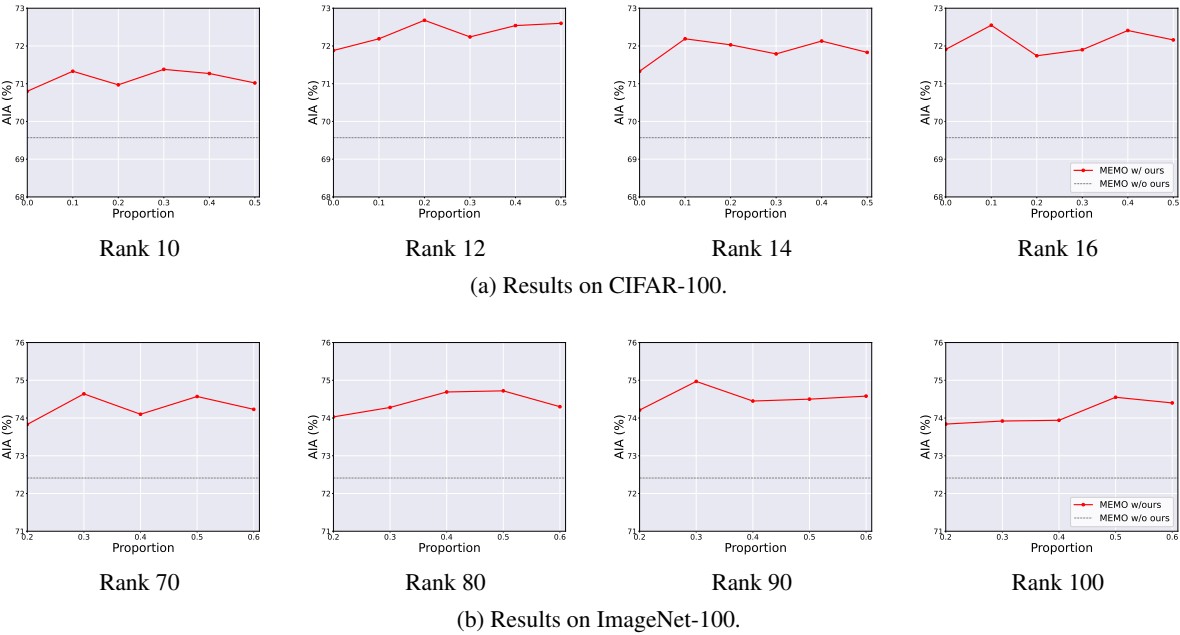

(a) Results on CIFAR-100.

(b) Results on ImageNet-100.

*Figure 3.* Results of different ranks $R$ and proportion $\epsilon$ on average accuracy (Avg %) with our methods integrated into MEMO. (a) CIFAR-100, (b) ImageNet-100.

*Table 6.* Classification accuracy on CIFAR-100 processed with different compression methods. "Accu." means the accuracy (%), "Comp." means the compressed ratio. "Down Sample" means setting the downsampling rate to 2.56, and "CIM" means keeping the 16*16 centre region as the original image, and the other parts of the image with a downsampling rate of 4.

| Method | Upper Bound | Down Smaple | CIM | TDMER | |
| --- | --- | --- | --- | --- | --- |
| | | | | 12 | 16 |
| **Accu.** | 72.3 | 44.1 | 66.9 | 67.8 | 69.9 |
| **Comp.** | 1.0 | 0.39 | 0.56 | 0.26 | 0.35 |

ing set using various compression methods, then trained a 32-layer ResNet from scratch on the original (upper bound) and processed training sets, respectively. The original test set was used for classification evaluation. As shown in Tab. 6, the TD-based compression method achieves considerable compression efficiency, and the gap between its performance and the upper bound is small. This indicates that our method retains sufficient discriminative information, allowing it to perform comparably to the original training set.

**Sensitivity Analyses.** We conducted experiments with different ranks and $\epsilon$, which controls the proportion of the raw and compressed data. As shown in Fig. 3 (a), different $\epsilon$ yield varying performance on the CIFAR-100, with the best performance observed around $\epsilon$ of 0.1 or 0.2. Simi-

larly, Fig. 3 (b) presents the results on the ImageNet-100. We observe that ImageNet-100 exhibits relatively stable performance across different $\epsilon$. However, the model's performance improves with increasing $\epsilon$, requiring a larger $\epsilon$ to achieve optimal performance. We attribute the different performances of the two datasets to their complexity and diversity. Overall, our exemplar selection strategy effectively balances the quality and quantity of exemplars, thereby enhancing performance. This approach provides a flexible mechanism to adapt to different dataset characteristics, ensuring optimal memory use and maintaining high model performance.

## 5. Conclusion

In this work, we propose a data-friendly and model-friendly memory efficient replay method based on CAN-DECOMP/PARAFAC (CP) decomposition. This method effectively leverages the local correlation and low intrinsic dimensionality natural images exhibit, achieving high compression efficiency while preserving discriminative information. Additionally, we introduced an innovative exemplar selection strategy to balance the quality and quantity of examples further. Extensive experiments consistently demonstrate that our methods significantly improve replay-based methods. Overall, our work represents a novel attempt to integrate tensor methods with CIL. We expect that the introduction of tensor methods will provide valuable insights and solutions to the challenges faced by the CIL community.

## Impact Statement

This paper presents work whose goal is to advance the field of Machine Learning. There are many potential societal consequences of our work, none which we feel must be specifically highlighted here.

## Acknowledgements

This work is supported in part by the Natural Science Foundation of China (NSFC) under Grant 62203124 and 62406077. We also thank the anonymous reviewers for their helpful comments.

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

# A. Implementation Details

## A.1. Training Details

Our implementation is based on the deep learning library PyTorch (Paszke et al., 2019) and the class-incremental learning toolbox PyCIL (Zhou et al., 2023a). Following (Yan et al., 2021; Wang et al., 2022a; Zhou et al., 2022; Luo et al., 2023), we employ the same backbone, i.e., 18-layer ResNet (He et al., 2016) for ImageNet-100 and 32-layer ResNet for CIFAR-100, with a fully-connected layer as the classifier across all our experiments. We utilize the Stochastic Gradient Descent (SGD) optimizer for training with the following hyperparameters:

- **Base** stage: We used SGD with an initial learning rate of 0.1, momentum of 0.9, and weight decay of 0.0005. The training epoch for all datasets is set to 200 with a batch size of 128. The learning rate is scheduled to decay by 0.1 at epochs 60, 120, and 170.

- **Inc**remental stage: We maintain the training epoch at 170, with a learning rate and momentum that remain constant concerning the base stage. Yet the learning rate decays with factor 0.1 at epochs 80 and 120, and the weight decay is adjusted to 0.0002. The learning rate decayed one more time in epoch 150 for MEMO.

Note that all hyperparameters remained consistent for FOSTER, except that it utilized a cosine annealing strategy for learning rate decay. Besides, for all baseline methods, we kept other hyperparameters consistent with those in PyCIL (Zhou et al., 2023a). Regarding the parameter of the CP decomposition in the main results, Tab. 7 lists them.

## A.2. The Details of Decomposition

This study presents a memory-efficient replay method based on Tensor Decomposition (Kolda & Bader, 2009). Specifically, we intend to introduce the tensor decomposition method, factorizing the raw image (a *third*-order tensor) as a set of factor matrices or tensors, and with the hyperparameter called 'Rank' to balance the quantity and quality of the decomposition. Hence, we need to solve two problems, i.e., the selection of the decomposition method and the Rank. Here, we briefly introduce several common tensor decomposition methods (taking a *third*-order tensor as an example).

First is the CANDECOMP/PARAFAC (CP) (Carroll & Chang, 1970), which expresses a tensor as a sum of rank-one tensors, each represented by a column vector from corresponding factor matrices along each mode. For a *third*-order tensor $\mathcal{X} \in \mathbb{R}^{I \times J \times \times K}$, the CP decomposition decomposes $\mathcal{X}$ into a sum of $R$ rank-one factor tensors:

$$\mathcal{X} \approx \sum_{r=1}^{R} \mathbf{a}_r \circ \mathbf{b}_r \circ \mathbf{c}_r \tag{8}$$

where $\mathbf{a}_r \in \mathbb{R}^I$, $\mathbf{b}_r \in \mathbb{R}^J$, $\mathbf{c}_r \in \mathbb{R}^K$, $R$ is the rank, and '$\circ$' is the outer product. The advantage of CP decomposition is its low storage complexity (only requires $\mathcal{O}(dnr)$ parameters) and computational complexity. Tucker decomposition (Tucker, 1966), which generalizes CP by allowing factor matrices to have different sizes, factorizes the tensor as a core tensor multiplied by a set of factor matrices along each mode. Formally, we can express it as:

$$\mathcal{X} \approx \mathcal{G} \times_1 \mathbf{A} \times_2 \mathbf{B} \times_3 \mathbf{C} \tag{9}$$

where $\mathcal{G} \in \mathbb{R}^{R_1 \times R_2 \times R_3}$ is the core tensor, $\mathbf{A} \in \mathbb{R}^{I \times R_1}$, $\mathbf{B} \in \mathbb{R}^{J \times R_2}$, $\mathbf{C} \in \mathbb{R}^{K \times R_3}$ is the factor matrices, '$\times_n$' represent the $n$-mode product, $R_1, R_2, R_3$ is a set of rank. Compared with CP decomposition, Tucker Decomposition provides a more flexible way to represent the data but needs more parameters, i.e., $\mathcal{O}(dnr + r^d)$. Tensor-Train (TT) Decomposition (Oseledets, 2011) decomposes a tensor into a series of chained products of *third*-order core tensors, which requires $\mathcal{O}(dnr^2)$ parameters. It can be represented as:

$$\mathcal{X}(i, j, k) \approx \sum_{R_1, R_2} \mathcal{G}_1(i, r_1) \cdot \mathcal{G}_2(r_1, j, r_2) \cdot \mathcal{G}_3(r_2, k) \tag{10}$$

where $\mathcal{G}_1 \in \mathbb{R}^{I \times R_1}$, $\mathcal{G}_2 \in \mathbb{R}^{R_1 \times J \times R_2}$, $\mathcal{G}_3 \in \mathbb{R}^{R_2 \times K}$ is the core tensors, $R_1, R_2$ is the rank. TT decomposition is widely used for high-dimensional data compression. The last one is the Tensor-Ring (TR) Decomposition (Zhao et al., 2016). TR is

a generalization of the TT decomposition, which decomposes a tensor into a series of cyclic products of $third$-order core tensors with the same parameter complexity as TT decomposition. Formally, it can be expressed as:

$$\mathcal{X}(i,j,k) \approx \sum_{R_1,R_2,R_3} \mathcal{G}_1(r_3,i,r_1) \cdot \mathcal{G}_2(r_1,j,r_2) \cdot \mathcal{G}_3(r_2,k,r_3) \tag{11}$$

where $\mathcal{G}_1 \in \mathbb{R}^{R_3 \times I \times R_1}, \mathcal{G}_2 \in \mathbb{R}^{R_1 \times J \times R_2}, \mathcal{G}_3 \in \mathbb{R}^{R_2 \times K \times R_3}$ is the core tensor, $R_1, R_2, R_3$ is the rank.

All of these methods can be used for compression, but we mainly consider two points: storage complexity and universality. It is obvious that CP decomposition has the lowest storage complexity. About the universality, although Tucker decomposition, TT decomposition, and TR decomposition offer greater flexibility, for low-resolution datasets like CIFAR-10/100, the size of the components will approach the original size to maintain the accuracy of the decomposition, which is undesirable in our case. To ensure generality, we opt to use only CP decomposition.

### A.3. The Details of Memory Information

About the detailed memory information, for CIFAR-100, when $R = 12$, $\epsilon = 0.2$, $M = 2k$, according to Eq. 3 compression rate $\eta \approx 0.26$, which means we saved 400 original samples and about 6100 sets of sample factors. Similarly, for ImageNet-100, when $R = 80$, $\epsilon = 0.4$, $M = 2k$, the compression ratio $\eta \approx 0.24$, which means we saved 800 original samples and about 5000 sets of sample factors

## B. More Experimental Results

As an extension of replay-based methods (Rebuffi et al., 2017; Lopez-Paz & Ranzato, 2017; Wang et al., 2022a), memory-efficient methods (Caccia et al., 2020; Zhao et al., 2021; Wang et al., 2022b; Luo et al., 2023; Zhai et al., 2023) improve storage efficiency by compressing the original image. Ideally, such methods should be fully adaptable to baseline methods. However, as mentioned earlier, it is often challenging for them to balance model-friendliness with data-friendliness. For example, CIM (Luo et al., 2023) only works well on high-resolution datasets but struggles with low-resolution ones (e.g., 32x32 CIFAR-100). MAE-CIL (Zhai et al., 2023) heavily dependent on MAE (He et al., 2022) for the image reconstruction. Here we provide more results to demonstrate further that our method is data-friendly and model-friendly. Tab. 7 lists the details of the datasets.

*Table 7.* Details of the three datasets. $R$ is the rank of decomposition, $\tau$ is reconstruction error threshold, and $\epsilon$ is the proportion of raw samples between decomposition factors.

| Dataset | Classes | Training images | Test images | Size | $R$ | $\tau$ | $\epsilon$ |
|---|---|---|---|---|---|---|---|
| CIFAR-100 | 100 | 50000 | 10000 | 32x32 | 12 | 0.07 | 0.2 |
| Tiny-ImageNet | 200 | 100000 | 10000 | 64x64 | 24 | 0.07 | 0.2 |
| ImageNet-100 | 100 | 129395 | 5,000 | 224×224 | 80 | 0.08 | 0.4 |

### B.1. Results on Tiny-ImageNet

We evaluate our method on three datasets with low, medium, and high resolutions, respectively. In the main text, we provide results on CIFAR-100 and ImageNet-100 (low and high ones), and Tab. 8 presents results on Tiny-ImageNet ($M = 2k$, 10-tasks setting, i.e., 'Base 0 Inc 20'). As shown, our method consistently improves baseline performance across datasets of various resolutions.

*Table 8.* average accuracy (%) and last accuracy (%) of four baseline methods on Tiny-ImageNet, with or without our TDMER plugged-in.

| Method | iCaRL | | FOSET | | MEMO | | DER | |
|---|---|---|---|---|---|---|---|---|
| | Avg | Last | Avg | Last | Avg | Last | Avg | Last |
| **w/o ours** | 35.84 | 15.38 | 50.86 | 39.90 | 50.22 | 37.91 | 52.51 | 41.01 |
| **w ours** | 39.40 | 18.35 | 54.32 | 41.20 | 57.85 | 48.81 | 58.48 | 50.54 |

## B.2. Results on ImageNet-1k

Here we provide the numerical incremental performance of DER (Yan et al., 2021) on ImageNet-1k, Tab. 9 lists the incremental and average accuracy. Note that the parameter configurations of ImageNet-1k remain the same as ImageNet-100, except that we set the memory budget $M = 20k$.

*Table 9.* Incremental and average accuracy comparison of DER under ImageNet1000 Base0 Inc100 setting, the memory budget $M = 20k$.

| Method | Accuracy in each session (%) | | | | | | | | | | Avg |
|---|---|---|---|---|---|---|---|---|---|---|---|
| | 1 | 2 | 3 | 4 | 5 | 6 | 7 | 8 | 9 | Last | |
| DER | 83.16 | 78.64 | 74.21 | 71.44 | 68.13 | 65.80 | 62.90 | 61.21 | 59.84 | 58.30 | 68.36 |
| DER w ours | 83.24 | 77.55 | 74.89 | 72.84 | 71.23 | 68.57 | 66.12 | 64.97 | 63.76 | 62.98 | 70.62 |

## B.3. Combine With Vision Transformer Based Methods

In recent years, the Vision Transformer (ViT) (Dosovitskiy, 2020) has garnered significant attention within the computer vision community, with many approaches to Class-Incremental Learning (CIL) being based on ViT. DyTox (Douillard et al., 2022) is a pioneering work that explores the use of ViT in CIL, which dynamically expands Transformer tokens to allocate independent representations for each task, effectively addressing catastrophic forgetting. Here, we combine our method with DyTox, and the results are shown in Tab. 10. As demonstrated, our method consistently improves DyTox performance across all settings, achieving greater improvements compared to the ViT-based efficient replay method MAE-CIL (Zhai et al., 2023).

*Table 10.* Results on CIFAR-100 in average accuracy (%) and last accuracy (%) on 10-, 20- and 50-task scenarios.

| Method | N =10 | | N =20 | | N =50 | |
|---|---|---|---|---|---|---|
| | Avg | Last | Avg | Last | Avg | Last |
| **Dytox** | 77.10 | 64.53 | 76.57 | 62.44 | 75.45 | 58.76 |
| **MAE-CIL** | 79.12 | 68.40 | 78.76 | 65.22 | 76.95 | 63.12 |
| **Dytox w ours** | **80.21** | **70.56** | **81.41** | **70.68** | **80.51** | **66.80** |

## B.4. Comparing with Other Compression Methods.

**Supplementary to Tab. 6**. To evaluate the effectiveness of the TD-based compression method, we processed the training set of CIFAR-100 (Krizhevsky, 2009) with different compression methods and tested it with the original test set. Specifically, for downsampling (Zhao et al., 2021), it can be implemented using the $resize()$ function, "2.56" and "4" represent the different downsampling ratio. For CIM (Luo et al., 2023), we did not take a dynamic determination of the activation region. We directly selected a 16*16 region in the centre and down-sampled the rest, settling the downsampling ratio to 4. More detailed results are shown in Tab. 11.

| Method | Upper Bound | Down-Sample | | CIM | TDMER | | | |
|---|---|---|---|---|---|---|---|---|
| | | 2.56 | 4.0 | | 10 | 12 | 16 | 20 |
| **Accu.** | 72.3 | 44.1 | 33.1 | 66.9 | 66.1 | 67.8 | 69.9 | 70.8 |
| **Comp.** | 1.00 | 0.39 | 0.25 | 0.56 | 0.22 | 0.26 | 0.35 | 0.44 |

*Table 11.* Classification accuracy on CIFAR-100 processed with different compression methods. "Accu." means the accuracy (%), "Comp." means the compressed ratio.

## B.5. Reconstructed Image visualization

Fig. 4 illustrates 15 randomly selected images from the original dataset (ImageNet-100) and the reconstructed images under different rank parameters.

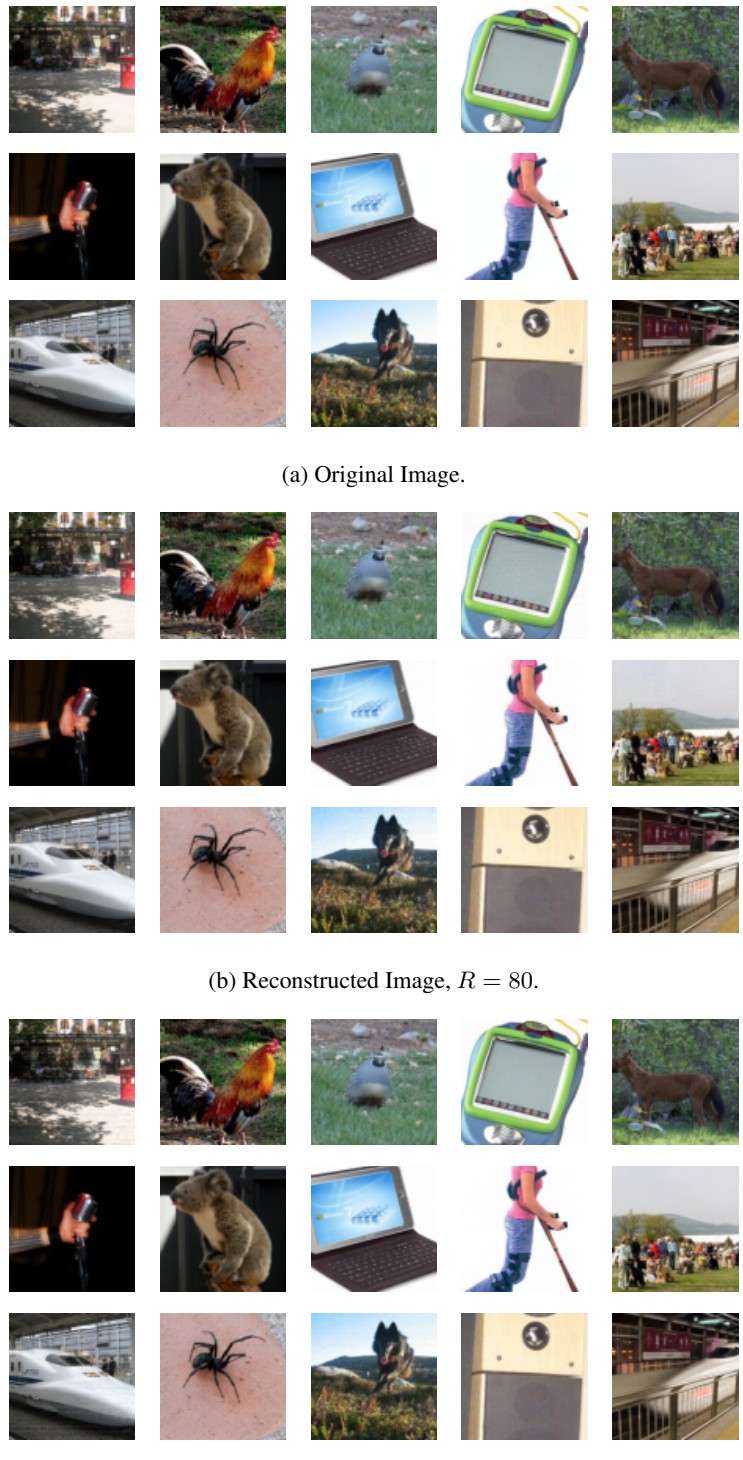

(a) Original Image.

(b) Reconstructed Image, $R = 80$.

(c) Reconstructed Image, $R = 100$.

*Figure 4.* ImageNet-100 image visualization. (a) Randomly selected 15 original images. (b), (c) Showing the reconstructed image at CP rank $R = 80$ and $R = 100$, respectively.

