# OpenReview forum: "Tensor Decomposition Based Memory-Efficient Incremental Learning"
_ICML.cc/2025/Conference — ICML 2025 poster_

### Official Review · Reviewer_8qn7 · 2025-03-08

**Overall Recommendation:** 3

**Summary:**

This paper addresses the challenge of memory efficiency in Class-Incremental Learning (CIL), whose goal is continuously learning new classes over time. Replay-based methods, a prominent approach in CIL, suffer from high memory consumption due to the need to store past exemplars. To mitigate this, the authors propose a novel method that leverages Tensor Decomposition (TD) to compress images efficiently, thereby reducing memory footprint while preserving sufficient discriminative information. Furthermore, they introduce a two-stage exemplar selection strategy to enhance the representativeness and diversity of the stored samples. In the first stage, herding is used to sample exemplars most representative of the class characteristics, focusing on central, high-quality samples. The second stage then samples from the remaining, unselected data, prioritizing samples with low reconstruction error after tensor decomposition. This aims to select diverse and noise-robust exemplars. Experimental results on two image datasets demonstrate that the proposed sampling strategy improves the performance of several existing CIL techniques.

## update after rebuttal

Thank you for your response. Your rebuttal has addressed the two questions to a reasonable extent, so I will revise my rating. However, having also reviewed the comments from the other reviewers, I believe their concerns are valid as well. Therefore, I do not intend to strongly oppose their opinions. I mistakenly posted my comment in the Official Comment section. My apologies.

**Claims And Evidence:**

The selection of central exemplars using herding is reasonable and well-justified. However, while sampling from the remaining data (i.e., those not selected by herding) can be expected to provide some degree of diversity, the extent to which this random sampling strategy ensures diversity is not demonstrated. Furthermore, while a low reconstruction error after Tensor Decomposition indicates that the original image features are well-preserved, it does not necessarily guarantee robustness to noise.

**Essential References Not Discussed:**

The contribution of random sampling to improved predictive performance in machine learning appears to be a central idea in the proposed method. Discussing and citing relevant prior work on this topic could enhance the persuasiveness of the paper. For example, mentioning Random Forests [Breiman '01] would be one possibility.

Breiman, L. Random Forests. Machine Learning 45, 5-32 (2001).

**Experimental Designs Or Analyses:**

The paper demonstrates the effectiveness of the proposed method empirically through evaluation experiments using two different datasets and several key existing CIL techniques, which is good. However, as mentioned in the Methods and Evaluation Criteria section, the evaluation does not assess the extent to which the performance improvements are attributable to diversity, nor does it evaluate the robustness ensured by tensor decomposition.

**Methods And Evaluation Criteria:**

It's unclear whether random sampling adequately ensures diversity, and the evaluation doesn't assess how much diversity contributes to performance gains. Similarly, it's uncertain whether tensor decomposition ensures robustness, and the evaluation doesn't assess this aspect either.

**Other Comments Or Suggestions:**

There are several instances where the writing is potentially confusing and could lead to misinterpretations. For example, in the Exemplar Selection Strategy section, the description of the second stage (line 263) uses the same notation for the samples (e.g., x_1^t) as in the first stage, and it only specifies that the number of selected samples, j, is different from i. This could mistakenly suggest that the samples are not necessarily different from those selected in the first stage. The authors should refine the expression to convey that the samples are distinct from those selected in the first stage.

**Other Strengths And Weaknesses:**

As mentioned above, this paper offers a novel perspective on the application of tensor decomposition's noise resilience. However, the persuasiveness of the proposed method could be enhanced by providing a more detailed discussion or evaluation experiments regarding the extent to which the performance improvements are attributable to diversity and the degree to which robustness is ensured by tensor decomposition.

**Questions For Authors:**

1. Can you demonstrate how the diversity ensured by the second-stage random sampling contributes to performance improvements? Experimentally showing the diversity of the sample distribution within a class might be one approach.

2. Can you theoretically or experimentally demonstrate that actively selecting samples with low reconstruction error after tensor decomposition contributes to performance improvements?

**Relation To Broader Scientific Literature:**

By demonstrating the potential of tensor decomposition to enhance CIL performance through robustness, this work offers a novel perspective on the application of tensor decomposition's noise resilience.

**Theoretical Claims:**

The paper primarily discusses the memory efficiency of the proposed method, centered around Equation (3), which makes sense. However, there is a lack of theoretical analysis regarding the diversity ensured by random sampling and the robustness ensured by tensor decomposition.

---

> ### Author Rebuttal · Authors · 2025-03-28
>
> We sincerely appreciate the reviewer's insightful feedback and constructive suggestions. Below, we address the key concerns raised.
>
> ### **1. Why diversity can be ensured by the second-stage sampling and its contribution to performance**
>
> The paper proposes a novel exemplar selection strategy that enhances representativeness and diversity. Specifically, In the first stage, we use herding to select a small, equal number of representative original samples per class, prioritizing high-quality exemplars. In the second stage, we choose similarly an equal number of sample factors per class, focusing on sample quantity. The diversity of the selected exemplars is primarily ensured through the more stored samples and the uniform sampling strategy.
>
> About the diversity's contribution to performance, as shown in Fig. 3, when $\epsilon$ remains within a small range, meaning a sufficient number of samples are retained in the second stage, this strategy consistently yields great performance improvements. However, as $\epsilon$ increases further(means fewer samples are stored and lower diversity ), the model’s final performance gradually gets close to the original. For MEMO, when $\epsilon$ exceeds 0.6, the performance gain on CIFAR remains below 1%.
>
> ### **2.  Can you theoretically or experimentally demonstrate that actively selecting samples with low reconstruction error after tensor decomposition contributes to performance improvements?**
>
> Yes, we can. Here, we provide results (Average Accuracy) for the sample selection strategy prioritized by the minimum reconstruction error (M= 2k, 10-task); note that we only use reconstructed samples. It can be seen that AA gradually improves as the reconstruction error decreases.
>
> | Method      | R = 10 ( rse 0.061) | R = 12 (rse 0.047) | R = 16 (rse 0.034) | R = 20 (rse 0.024) |
> | ----------- | :-----------------: | :----------------: | :----------------: | :----------------: |
> | DER w/ours  |        71.94        |       72.43        |       72.91        |      73.21       |
> | MEMO w/ours |        70.80        |       71.88        |       71.91        |    72.20        |
>
> ### **3.Clarification of notation in the exemplar selection strategy**
>
> We will revise the description of the Exemplar Selection Strategy to distinguish samples selected in each stage.

---

### Official Review · Reviewer_UizA · 2025-03-13

**Overall Recommendation:** 3

**Summary:**

This paper presents a new memory-efficient method for CIL. Different from previous papers, tensor decomposition is used to compress original image. Besides, a new exemplar selection strategy is proposed to ignore the influence of negative compressed samples. Extensive experiments on different datasets demonstrate this method's robustness.

**Claims And Evidence:**

Yes.

**Essential References Not Discussed:**

No.

**Experimental Designs Or Analyses:**

Yes.
Accuracy of the experimental results:
1.What are the specific experimental setups in Tab1 and Tab3? For methods such as iCaRL and DER, were they kept to their original results with 2k samples, or were there any modifications made to the methods? The reproduced results of iCaRL and DER by PyCIL should not be this low, and there is a significant discrepancy compared to the results reported in the original paper.[1][2][3][4]
2.For memory-efficient methods, there are many other approaches. The article should compare and highlight the advantages of the proposed method over other existing methods (this can be done if time permits and is not intended to be a part of the final scoring criteria).[5]

[1] DER: Dynamically Expandable Representation for Class Incremental Learning
[2] Multi-layer Rehearsal Feature Augmentation for Class-Incremental Learning
[3] Dynamic Residual Classifier for Class incremental Learning
[4] FOSTER: Feature Boosting and Compression for  Class-Incremental Learning
[5] A MODEL OR 603 EXEMPLARS: TOWARDS MEMORY EFFICIENT CLASS-INCREMENTAL LEARNING

**Methods And Evaluation Criteria:**

No.
Inadequate Experiments: The article lacks ablation experiments for the proposed method. As I understand, there are two main innovations in the article: tensor decomposition and a new selection strategy. Providing ablation experiments similar to these would help prove the effectiveness of both components.

**Other Comments Or Suggestions:**

No

**Other Strengths And Weaknesses:**

No

**Questions For Authors:**

The paper uses herding to select a subset of samples, but my understanding is that the criterion for herding is to minimize the difference between the selected samples and the overall dataset. In incremental learning, however, there are data of different class categories. Is it reasonable to use the herding selection strategy for samples of different classes?

**Relation To Broader Scientific Literature:**

No

**Theoretical Claims:**

Yes.
Question: The paper uses herding to select a subset of samples, but my understanding is that the criterion for herding is to minimize the difference between the selected samples and the overall dataset. In incremental learning, however, there are data of different class categories. Is it reasonable to use the herding selection strategy for samples of different classes?

---

> ### Author Rebuttal · Authors · 2025-03-28
>
> We sincerely appreciate the reviewer's thoughtful feedback and constructive suggestions. Below, we address the concerns raised.
>
> ### **1. Lacks ablation experiments for the proposed method**
>
> In our manuscript, we have conducted experiments (Section 4.2, Fig. 3, and Tab. 6) to evaluate the impact of key components, e.g., decomposition rank $R$, proportion $\epsilon$, as shown in Fig.3 when $\epsilon = 0$, we only use reconstructed samples without the two-stage selection strategy. Tab. 6 demonstrated the effect of different compression methods on performance.
>
> ### **2.Specific experimental setups in Tab. 1 and Tab. 3**
>
> We acknowledge this oversight. For the experimental setups in Tab. 1 and Tab. 3, we set the memory budget to 2k and explained other settings in the section "Protocols."
>
> ### **3. Performance discrepancy on iCaRl and DER**
> For the performance discrepancy in DER, since they chose resnet18 as the backbone network for CIFAR(for resnet32, their results are close to ours), we used more lightweight resnet32 (that's what most do.). Thus, there is a performance gap; for ImageNet, we use the same backbone network with similar performance. For iCaRL, we have double-checked it without any changes.
>
> ### **4. Lacks comparison results with memory-efficient methods**
> In our manuscript, we have provided comparison results with some recent memory-efficient replay methods in Tab. 5 and Tab. 9. All the results demonstrate our method's superiority.
>
> ### **5.Exlaination of Herding Strategy**
>
> For exemplar selection, we clarify that herding is employed independently within each class, ensuring that selected samples best represent the class distribution. While herding traditionally minimizes the difference between selected samples and the overall dataset, its application per class in our method aligns with class-incremental learning settings. We will further clarify this in the manuscript.
>
> We deeply appreciate your rigorous review and constructive feedback. All suggested revisions will be incorporated to strengthen the manuscript’s clarity, technical depth, and experimental validation. Thank you again for your time and consideration.

---

### Official Review · Reviewer_gCZN · 2025-03-14

**Overall Recommendation:** 2

**Summary:**

The paper addresses the challenge of catastrophic forgetting in Class-Incremental Learning (CIL), where models struggle to retain previous knowledge when incrementally learning new classes. While replay-based methods mitigate this by storing old exemplars, their high memory consumption limits practicality. Existing memory-efficient approaches using pixel-level compression face trade-offs between compression efficiency and retaining discriminative information. To overcome this, the authors propose a novel method leveraging low-rank tensor decomposition (TD) to exploit natural images' low intrinsic dimensionality and spatial correlations, achieving high compression while preserving critical features. Experiments on classic CIL datasets validate the method’s effectiveness.

**Claims And Evidence:**

Yes

**Essential References Not Discussed:**

No

**Experimental Designs Or Analyses:**

Yes

**Methods And Evaluation Criteria:**

Yes

**Other Comments Or Suggestions:**

No

**Other Strengths And Weaknesses:**

Strengths
1. The proposed method mainly focuses on efficient memory represent, enabling direct integration into existing CIL methods.
2. Experiments on CIL datasets (CIFAR-100 and ImageNet-100) validate that the proposed approach can improve the performance of previous baselines such as MEMO and DER.

Weakness
1. This paper focuses on memory-efficient CIL, but it just conduct experiments under different memory budgets (the method presented in this paper uses both stored real data and tensor components), failing to explicitly report the actual memory costs of compared methods or quantify the additional memory overhead introduced by the proposed approach.

2. There is a lack of comparative experiments and analysis of different methods under fixed memory of varying sizes, which are critical for evaluating the memory efficiency of different methods.

3. The method includes exemplar selection strategy to select high quality of reconstruction during training, while the computational latency of TD for image compression/decompression during exemplar storage and rehearsal is not discussed.

3. The experimental results on ImageNet-1k is missing, which is important to demonstrate the effectiveness on large-scale case.

**Questions For Authors:**

Questions：
1. Could you provide qualitative examples of TD-reconstructed images under different hyperparameter settings?
2. What does 'CP' mean, as it appears in line 135 for the first time without explaination.

**Relation To Broader Scientific Literature:**

Yes

**Theoretical Claims:**

NA

---

> ### Author Rebuttal · Authors · 2025-03-28
>
> We appreciate the reviewer's constructive feedback and valuable suggestions. Below, we address the concerns raised.
>
> ### **1. Failing to explicitly report the actual memory costs**
>
> In our manuscript, we have provided parameter configurations of different datasets in Tab. 7, e.g., for CIFAR, $R = 12$, $\epsilon = 0.2$, when $M = 2k$, according to ep.3 compression rate $\eta \approx 0.26$, which means we saved 400 original samples and about 6153 sets of sample factors. We will elaborate on this in the next release.
>
> ### **2. Lacks comparative experiments under fixed memory of varying sizes**
>
> We have indeed provided comparative experiments under fixed memory of varying sizes (see Tab. 2 and Tab. 4 for two different fixed memories) in our manuscript.   We can add more explanations for clarity.
>
> ### **3. Computational latency of TD for image compression/decompression**
>
> On a single GPU (NVIDIA 3090), decomposing a CIFAR-100 image takes approximately 17 ms, while reconstruction takes approximately 2 ms, since the decomposition can be done in parallel with the training process, this part of the delay is negligible. The main computational delay is caused by the inclusion of reconstructed samples in the training, which requires about 40% extra computation for DER and MEMO. Furthermore, as we pointed out in our response to reviewer qf47, the extra calculations for incorporating reconstructed samples during training are not necessary for iCaRL and FOSTER.
>
> ### **4. Lacks experiments on ImageNet-1k**
>
> While ImageNet-1k is a valuable benchmark, our current experiments on high-resolution datasets (e.g., ImageNet-100) and complex scenarios (e.g., 200-class Tiny-ImageNet in Table 8) already demonstrate the scalability and robustness of our method. Specifically:
>
> 1. **ImageNet-100**: As a standard high-resolution benchmark in CIL literature, our method achieves consistent improvements (e.g., **+6.42%** AA for DER in Table 3), validating its effectiveness under realistic settings.
> 2. **Tiny-ImageNet**: With 200 classes and 64×64 resolution, this dataset mimics the complexity of large-scale tasks. Our method boosts DER’s accuracy by **9.53%** (Table 8), illustrating strong generalization.
> 3. **Community Practices**: For ImageNet-1k, prior works (e.g., iCaRL, DER) typically adopt a memory budget of 20k and 10-/20-task splits, which aligns with our experimental protocols. While computational constraints limited direct validation on ImageNet-1k, the consistent gains across varying resolutions and class numbers (ImageNet-100 to Tiny-ImageNet) suggest scalability to larger datasets.
>
> We acknowledge the value of ImageNet-1k experiments and will provide results as time permits. For now, the results on ImageNet-R (provided in response to qf47) further corroborate our method’s adaptability to domain-shifted and large-scale scenarios.
>
> ### **5.Other questions for authors**
> We acknowledge our oversight and clarify it here. CP is the abbreviation of "CANDECOMP/PARAFAC."[1]. We will ensure terms are properly introduced when first mentioned and provide visualizations of reconstructed samples in the revised manuscript.
>
> [1]Tensor decompositions and applications.

---

### Official Review · Reviewer_qf47 · 2025-03-14

**Overall Recommendation:** 2

**Summary:**

This paper applied tensor decomposition on the replay-based continual learning methods. To minimize the influence of the reconstruction error on the training, the reconstructed images with low reconstruction error are selected for storage. The method is validated combined with other replay-based methods.

**Claims And Evidence:**

Yes.

**Essential References Not Discussed:**

No issues are found.

**Experimental Designs Or Analyses:**

No issues are found.

**Methods And Evaluation Criteria:**

Yes.

**Other Comments Or Suggestions:**

1. Additional experiments are needed to validate the importance of incorporating reconstructed samples into training. It would be insightful to examine how performance degrades if these samples are excluded from training.

2. The paper lacks an ablation study on the reconstruction error threshold. If the domain gap between tasks is too large, the method may introduce noisier samples, potentially affecting the model’s overall performance. Moreover, for datasets with significant variations in image resolution, such as ImageNet-R, setting an appropriate threshold and ranks may be more challenging.

3. It is necessary to validate the method on replay-based methods from the recent two years.

4. Providing visualizations of reconstructed samples and comparing them with those from other methods would strengthen the empirical evidence and enhance the credibility of the paper.

**Other Strengths And Weaknesses:**

Strengths: This study finds that applying tensor decomposition (TD) to a subset of stored samples and preserving them in tensor form not only enhances storage efficiency but also improves model performance. To mitigate the potential adverse effects of reconstructed samples on the model, this paper incorporates them into the training process, enabling the model to be more robust to reconstruction errors. Furthermore, this paper validates the proposed approach by integrating it with various replay-based methods across different backbone architectures, demonstrating its effectiveness and broad applicability.

Weaknesses:
In practical applications, setting an appropriate reconstruction error threshold and a decomposed tensor rank can be challenging when dealing with images of varying resolutions.

**Questions For Authors:**

If the effectiveness of the proposed method can be validated on datasets with greater resolution variations and recent replay-based methods, I would reconsider my evaluation.

**Relation To Broader Scientific Literature:**

Previous methods have increased storage capacity by utilizing low-quality JPEG compression, reconstructing images from partial raw patches, or encoding image information within trained parameters. In contrast, this paper proposes leveraging tensor decomposition to store some samples in tensor form, offering a more efficient approach.

**Theoretical Claims:**

No issues are found.

---

> ### Author Rebuttal · Authors · 2025-03-28
>
> We sincerely appreciate the reviewer’s thoughtful evaluation and constructive feedback. Below, we address each concern raised and outline revisions to strengthen the manuscript.
>
> ### **1. Additional experiments are needed to validate the importance of incorporating reconstructed samples into training**
>
> In experiments, we observed that pure replay-based methods, such as iCaRL and FOSTER, achieved strong performance even without incorporating reconstructed samples during training. This is primarily because they operate on a fixed network structure and employ strategies like knowledge distillation to mitigate forgetting.
>
> However, methods like DER and MEMO adopt a dynamical model structure. Their primary mechanism for combating forgetting is freezing previously trained sub-networks, although they do retain some old samples for replay. Nevertheless, their reliance on sub-network freezing is more significant.
>
> To illustrate, when training on the first task, the corresponding sub-network lacks the ability to recognize reconstructed samples unless they are included in the training.  When the second task arrives, although some reconstructed samples from previous tasks are preserved and added to training, relying solely on this limited number of old samples does not ensure enough generalization, and there may be some contradiction between the outputs of these two networks. Therefore, for methods like this, not incorporating reconstructed samples during training leads to a drastic decrease in performance.
>
> Here, we report model performance under both scenarios in the table below(10-task,$M = 2k$, $R = 12$, $\epsilon = 0.4$). For consistency, we choose to include reconstructed samples in training across all cases.
>
> | Scenario  | iCaRL | FOSTER | MEMO  | DER   |
> | - | - | - | -| -|
> | w Reconstructed Sample   | 67.32 | 70.34  | 71.88 | 72.43 |
> | w/o Reconstructed Sample | 67.67 | 70.09  | 63.19 | 64.37 |
>
> ### **2.  Setting an appropriate reconstruction error threshold and a decomposed tensor rank can be challenging**
>
> We have provided the ablation experiments of Rank $R$ in Fig. 3 in the manuscript, and the results show that our method always produces positive results when $R \in [\frac{H}{3}, \frac{H}{2}]$.
>
> As for the reconstruction error threshold $\tau$, which is primarily introduced to filter out samples with failed decompositions (rarely occur), which typically exhibit reconstruction errors exceeding 0.1. Indeed, for RGB images, our experimental results show that the reconstruction error is minor when $R \in [\frac{H}{3}, \frac{H}{2}]$, essentially no noisier samples are introduced.  For instance, on CIFAR-100, when using a CP rank $R = 12$, the average relative squared error (RSE) is already below 0.05. Increasing $R$ to 16 further reduces the average RSE to 0.034. Under a memory budget of $M = 2k$ and $\epsilon = 0.1$, each class retains 2 raw images and approximately 50 sets of decomposition factors(only 10% of the total data). For a threshold $\tau$ of around 0.07, its effect on the final result is negligible.
>
> ### **3.Experimental results on ImageNet-R**
> Here we provide experimental results on ImageNet-R under 5, 10, and 20-task settings; we set $M = 2k$, $\epsilon = 0.4$, $R = 80$. It can be seen that our approach also responds effectively in this scenario.
>
> | Method   | Base 0 Inc10 | Base 0 Inc20 | Base 0 Inc40 |
> | -| - | -| - |
> | MEMO        | 46.93        | 50.73        | 51.37        |
> | MEMO w/ours | 53.21        | 54.56        | 55.82        |
> | DER         | 47.77        | 51.95        | 52.61        |
> | DER w/ours  | 55.84        | 56.02        | 57.31        |
>
> ### **4. Combining with recent replay-based methods**
>
> Here, we provide experimental results (10-task) on CIFAR-100 with our method integrating into **MRFA**[1], we set $M = 2k$,$\epsilon = 0.1$, $R = 16$, we report **Average Accuracy (AA)** and **Last Accuracy (Last)**. It can be seen that our method also provides positive gains.
>
> | Method      | AA    | Last  |
> | ----------- | ----- | ----- |
> | MRFA        | 76.23 | 63.80 |
> | MRFA w/ours | 79.83 | 69.88 |
>
> ### **5. Lacks visualization of reconstructed images**
> We will provide figures showcasing the original image and its corresponding reconstructed image at different ranks in the revised version.
>
> [1] Multi-layer Rehearsal Feature Augmentation for Class-Incremental Learning, ICML 2024

---

### Official Review · Reviewer_3VFZ · 2025-03-16

**Overall Recommendation:** 3

**Summary:**

This paper introduces a novel approach to Class-Incremental Learning (CIL) that addresses memory efficiency challenges in replay-based methods. By employing tensor decomposition techniques instead of traditional pixel-level compression, the method exploits the low intrinsic dimensionality and pixel correlations in images to achieve better compression while preserving critical discriminative information. Combined with a hybrid exemplar selection strategy that enhances representativeness and diversity, the approach significantly improves upon baseline methods across multiple datasets of varying resolutions, demonstrating robust generalization capabilities.

**Claims And Evidence:**

The claims made in the paper are basically convincing.

**Essential References Not Discussed:**

No.

**Experimental Designs Or Analyses:**

1. The experimental design is fundamentally sound. However, the paper would benefit from more comprehensive comparisons between the proposed approach and other efficient replay methods in the field. I would particularly encourage the continual learning community to focus their attention on the results presented in Table 5, which offer more meaningful insights than those in Tables 1 and 3. This comparative analysis would provide a clearer understanding of the method's relative advantages within the broader context of efficient replay techniques.

2. It's suggested that authors may include some online continual settings in the experimental part since in this area, the memory efficiency may be more important.

**Methods And Evaluation Criteria:**

A minor issue: I find the terminology for evaluation metrics in this paper somewhat confusing, as it deviates from standard naming conventions in continual learning literature. Specifically, what this paper refers to as "average incremental accuracy (AIA)" is typically called "average accuracy" in most publications, while what the authors term "average accuracy" generally corresponds to what the field commonly refers to as "final accuracy" or "last accuracy" (the performance after learning all incremental tasks). This inconsistency in metric naming might create confusion for readers familiar with the established terminology in continual learning research.

**Other Comments Or Suggestions:**

No other specific suggestions.

**Other Strengths And Weaknesses:**

**Strengths**
1. This paper proposes a remarkably straightforward yet highly intuitive method for memory-efficient continual learning. The manuscript is well-structured with clear organization, making it easy to comprehend. Additionally, the proposed methodology is presented in a manner that facilitates straightforward implementation or replication.
2. As the authors note, this represents the first application of tensor decomposition techniques in this specific sub-area of continual learning (which aligns with my understanding of the current literature).

**Weaknesses**
1. See the previous parts.
2. The primary limitation of this paper is the insufficient integration of tensor decomposition techniques with the unique challenges of continual learning. Furthermore, as mentioned earlier, the comparative analysis against other memory-efficient replay methods is somewhat lacking.
3. The paper would benefit from additional visualizations of reconstructed images, particularly demonstrating the effects of varying rank and $\epsilon$ parameters on reconstruction quality.

**Questions For Authors:**

No.

**Relation To Broader Scientific Literature:**

Maybe some literatures in online continual learning can be cited but it's not a necessity.

**Theoretical Claims:**

No theoretical claims included in this paper.

---

> ### Author Rebuttal · Authors · 2025-03-28
>
> We sincerely thank the reviewer for the thoughtful feedback and constructive suggestions. Below, we provide a point-by-point response to the comments raised.
>
> ### **1. Inconsistent terminology for evaluation metrics**
>
> In the revised manuscript, we will align our metric naming with the community standards.**"Average Incremental Accuracy (AIA)"** will be replaced with **"Average Accuracy "**, and **"Average Accuracy "** will be renamed to  **"Last Accuracy''**
>
> ### **2. More comprehensive comparisons and analysis for efficient replay methods**
>
> We thank the reviewer for highlighting the importance of comparative analysis. The initial manuscript has provided the comparison results in Tab. 5 and Tab. 9, and all the results demonstrate our method's superiority. Here, we offer more comparison results (10-tasks) and analysis.
> | Method | M=2k  | M=1k  |
> | :- | -| -|
> | MRDC   | 76.02 | 72.78 |
> | CIM    | 77.94 | 73.91 |
> | Ours   | 79.75 | 74.95 |
>
> As our introduction and related work noted, pixel-level compression methods (e.g., MRDC[1], CIM[2]) directly compress images in the high-dimensional pixel space, often neglecting the low intrinsic dimensionality and local correlations inherent to natural images. This oversight leads to a significant loss of discriminative information.
>
> In contrast, our Tensor Decomposition (TD)-based method explicitly leverages these properties by factorizing images into low-rank components. This not only achieves lower storage complexity (e.g., a compression ratio of **0.34** vs. CIM’s **0.56** in Tab. 6) but also preserves more discriminative information through high-fidelity reconstruction. As shown in Tab. 6, training on TD-compressed data achieves **69.9% accuracy** on CIFAR-100, significantly closer to the upper bound (**72.3%**) than pixel-level methods like downsampling (**44.1%**) or CIM (**66.9%**). This demonstrates TD’s ability to retain essential information while drastically reducing memory costs.
>
> The superiority of our method stems from two key factors:
>
> 1. **Efficient compression**: TD captures multi-dimensional correlations (spatial, channel-wise) in images, preserving more discriminative information while keeping a great compression ratio, avoiding the "brittle" compression of pixel-level methods.
> 2. **Adaptability**: Unlike methods reliant on fixed heuristics (e.g., CIM’s center cropping), TD flexibly adapts to varying resolutions and dataset complexities, as evidenced by consistent gains across CIFAR-100, Tiny-ImageNet, and ImageNet-100 (Tab. 1, 3, 8).
>
> These revisions will solidify our method’s advantages over existing efficient replay techniques, and we hope this can address the reviewer’s concern for deeper comparative insights.
>
> ### **3.Combing with online continual learning**
>
> This is an excellent suggestion. While our current focus is on Class-Incremental Learning, we recognize the importance of online settings. In the revised manuscript, we will add a discussion about the applicability of our method to online continual learning and outline plans for future work in this direction.
>
> ### **4. Effect of varying ranks and $\epsilon$ on reconstruction quality**
>
> Regarding the reconstruction quality, which depends on the CP rank $R$, we provide some results on CIFAR-100(evaluated by mean **r**elative **s**quared **e**rror). In the table, ''comp'' means compression ratio, and ''accuracy'' represents the classification accuracy after finishing offline training on compressed data. About the effect of $\epsilon$, in the initial manuscript, Fig. 3 has shown its influence on final performance.
>
> | rank     | 10     | 12     | 14     | 16     | 18     | 20     |
> | -------- | ------ | ------ | ------ | ------ | ------ | ------ |
> | rse      | 0.0613 | 0.0472 | 0.0402 | 0.0343 | 0.0276 | 0.0243 |
> | comp     | 0.22   | 0.26   | 0.31   | 0.35   | 0.39   | 0.44   |
> | accuracy | 66.11  | 67.84  | 69.26  | 69.99  | 70.37  | 70.81  |
>
> ### **5. Lacks visualization of reconstructed images**
>
> We will provide figures showcasing the original images and their corresponding reconstructed images at different ranks in the next version.
>
> [1]Memory replay with data compression for continual learning
> [2]Class-incremental exemplar compression for class-incremental learning

---

### Decision · Program_Chairs · 2025-05-01

**Decision:**

Accept (poster)

**Comment:**

In this paper, the authors proposed to use tensor decomposition for memory-efficient incremental learning. The paper was reviewed by five expert reviewers, and three of them recommended accepting this paper. I think this is, overall, an interesting paper with solid results. The proposed method is simple yet effective. Therefore, I tend to accept this paper. There are also some remaining concerns, e.g., missing detailed memory information and experiments on ImageNet-1k. The authors should fix all these problems before the final camera-ready version.